# Molecular Targets of Natural Compounds with Anti-Cancer Properties

**DOI:** 10.3390/ijms222413659

**Published:** 2021-12-20

**Authors:** Małgorzata Kubczak, Aleksandra Szustka, Małgorzata Rogalińska

**Affiliations:** 1Department of General Biophysics, Faculty of Biology and Environmental Protection, University of Lodz, Pomorska 141/143, 90-237 Łódź, Poland; malgorzata.kubczak@biol.uni.lodz.pl; 2Department of Cytobiochemistry, Faculty of Biology and Environmental Protection, University of Lodz, Pomorska 141/143, 90-237 Łódź, Poland; aleksandra.szustka@biol.uni.lodz.pl

**Keywords:** natural anti-cancer agents, curcumin, graviola, resveratrol, quercetin, lycopene

## Abstract

Cancer is the second leading cause of death in humans. Despite rapid developments in diagnostic methods and therapies, metastasis and resistance to administrated drugs are the main obstacles to successful treatment. Therefore, the main challenge should be the diagnosis and design of optimal therapeutic strategies for patients to increase their chances of responding positively to treatment and increase their life expectancy. In many types of cancer, a deregulation of multiple pathways has been found. This includes disturbances in cellular metabolism, cell cycle, apoptosis, angiogenesis, or epigenetic modifications. Additionally, signals received from the microenvironment may significantly contribute to cancer development. Chemical agents obtained from natural sources seem to be very attractive alternatives to synthetic compounds. They can exhibit similar anti-cancer potential, usually with reduced side effects. It was reported that natural compounds obtained from fruits and vegetables, e.g., polyphenols, flavonoids, stilbenes, carotenoids and acetogenins, might be effective against cancer cells in vitro and in vivo. Several published results indicate the activity of natural compounds on protein expression by its influence on transcription factors. They could also be involved in alterations in cellular response, cell signaling and epigenetic modifications. Such natural components could be used in our diet for anti-cancer protection. In this review, the activities of natural compounds, including anti-cancer properties, are described. The influence of natural agents on cancer cell metabolism, proliferation, signal transduction and epigenetic modifications is highlighted.

## 1. Introduction

Cancer is a major cause of morbidity, with millions of new cases every year. It is commonly known that cancer displays a high heterogeneity. In 2011, Hanahan and Weinberg described ten different hallmarks of cancer to allow a better understanding of the diversity of this type of disease. Cancer transformation can be a cause of disruptions in many different cellular metabolic pathways [1]. Therefore, discovering new, effective drugs against cancer seems to be an important challenge for scientists. New anti-cancer agents have exhibited their activity in vitro, as well as in clinical trials, and thus there is a high probability, but not a guarantee, that these drugs will be effective in vivo [2]. Despite the growing number of advanced treatment regimens and better access to new generations of anti-cancer agents, many types of cancer in advanced phases of development remain incurable. Some chemotherapeutics affect cancer cells but do not destroy cancer stem cells, leading to disease relapse and refractoriness to treatment [3,4]. Multidrug resistance may be a consequence of therapy application and could undoubtedly lead to serious obstructions in cancer therapy. Moreover, the increasing number of experiments indicate that microenvironmental components may influence a cell’s response to therapy [5].

Chemically synthesized anti-cancer agents are the largest group of drugs approved for cancer treatment around the world. A number of them are based on structures present in natural compounds obtained from plants.

Polyphenols represent a group of natural compounds originating mainly from plants. These molecules contain aromatic rings, as well as one or more hydroxyl groups (Figure 1). Polyphenols are divided into flavonoids, chalcones, stilbenes, curcuminoids and phenolic acids [6].

Usually, polyphenols found in food can be identified as esters, glycosides or a kind of polymers. Polyphenols derived from food (e.g., from dark chocolate, grapes, tea, apples, coffee) are absorbed in the small intestine. Several studies in vivo and in vitro confirm the anti-oxidative, anti-inflammatory, or metabolic-related activity of polyphenols [7]. Moreover, cocoa and dark chocolate can stimulate microbiota, leading to anti-inflammatory protection. It is also probable that a particular amount of polyphenols consumed in a diet can serve as anti-cancer protection. Interestingly, the results of studies show that another ferulic acid derivative is able to inhibit induction of NF-κB expression, which inhibits inflammation [8].

Generally, there is an increasing rank of molecular targets for natural or chemically synthesized therapeutics: signaling pathways, immunological system elements, cancer cell metabolism, cell cycle, angiogenesis, apoptosis and even epigenetic modifications [9,10,11,12,13,14,15]. Unfortunately, despite their activity, serious adverse reactions can limit their usage in cancer treatment [3].

Natural compounds isolated from plant extracts seem to be a very attractive alternative to conventional treatments. Natural compounds offer structural similarities between chemical and natural compounds, low toxicity and a lack of adverse reactions. However, experimental results demonstrate that reaching therapeutic concentrations of natural compounds in blood and tissue can be very difficult. Natural agents’ rapid biotransformation in the liver can be important in obtaining the proper curative concentrations of agents [16]. The best solution to deliver the proper amount of compounds directly to cancer cells is to connect and transport such agents inside liposomes or dendrimers [17,18]. It must be stated that these therapeutic approaches require further investigation to exclude any potent toxicity of such conjugates toward normal cells. A more precise monitoring of natural agents’ influence on the immunological system is also needed.

In this review, we summarize how natural compounds may affect cancer cells by targeting transcription factors, the microenvironment, epigenetic modifications, cancer stem cells and cancer cell metabolism.

## 2. Curcumin

Curcumin (Figure 1A) is a plant compound [1,7-bis(4-hydroxy-3-methoxyphenyl)-1,6heptadiene-3,5-dione] derived from turmeric *Curcuma zedoaria* (>100 µg/g), *Curcuma longa* (1–2 µg/g) and *Curcuma aromatica* (0.1 µg/g), a commonly used spice in Asia [19,20,21]. Curcumin can show keto–enol tautomerization depending on pH, i.e., at pH < 7 mainly the keto-form is present, but at pH > 7, the enol form occurs [22]. Curcumin can have three pKa values: the first (pKa 7.7–8.5), the second (pKa 8.5–10.4) and the third (pKa 9.5–10.7). Moreover, in PH 1–6 it is chemically stable and insoluble in water [19]. Therefore, when a person experiences some metabolic alteration caused by ketosis, this might be a reason for progressive increase in ketone bodies in the human body during the development of diabetes. These pH changes can also change curcumin structure and reactivity in the human body, even from curcumin included in the human diet.

Moreover, in alkaline media, curcumin can degrade into [trans-6-(4′-hydroxy-3′-methoxyphenyl)-2,4-dioxo-5-hexenal] and also vanillin, ferulic acid and feruloyl methane. Degradation of curcumin, mainly by oxidation, changes curcumin structure and properties, affecting pharmacokinetic and pharmacodynamic activity [20]. Many therapeutic properties have been assigned to curcumin, i.e., anti-inflammatory (arthritis, psoriasis), anti-tumor (colorectal cancer), anti-bacterial, anti-protozoal, anti-viral, wound-healing and chemoprotective activity. Curcumin can be helpful in the treatment of Alzheimer, neuronal disorders (headaches, depression), fever, malaria, mental health problems, worms, diabetes, cardiometabolic problems and also disorders in kidney function or water retention [19,20,23,24,25,26,27,28,29,30,31]. In Asian cultures, curcumin has been used in wound healing and for curing digestive system disorders for more than 4000 years. The first reports confirming curcumin activity appeared in the 1970s. The interest concerning this compound increases every year. More than 1000 publications about the spice’s properties are available (PubMed). Scientists believe that it could be an attractive anti-cancer agent because of its pleiotropic activity on cancer cells and simultaneously low cytotoxicity on normal cells. Since it was found that plant compounds display chemoprotective functions, plant extracts have been used in combination therapies. Unfortunately, a big disadvantage of many plant compounds, including curcumin, is their low bioavailability and absorption. Curcumin absorbs poorly from the digestive system to the blood after oral administration because of the hydrophobic character of this agent. Most administered doses are biotransformed in the liver to inactive hydrophilic derivatives. Only a small volume of active curcumin goes into blood circulation to reach the target tissues. First-phase trials indicate that curcumin concentration is barely detectable in plasma, and its therapeutic effects were observed only in particular parts of the digestive system (mainly in the large intestine) [16]. Curcumin encapsulation can be a solution for improving the stability, bioavailability and hydrophilicity of the agent. It can also be helpful in preventing side effects when using a high dose of the compound. This can be important in preventing curcumin’s low activity in an alkalic environment, as well as in contact with metallic reactive ions or ascorbic acid residues. Such encapsulation can be produced by amorphous or crystalline-related polymers, hydrogels, lipids, nano-carriers or matrices consisting of special ionic combining structures [32,33,34]. The other way to increase curcumin bioavailability in peripheral blood is to attach molecules inside liposomes or to connect them with nanoparticles, e.g., dendrimers or gold molecules.

There are several known curcumin derivatives (e.g., demethoxycurcumin, tetrahydrocurcumin, hexahydrocurcumin, octahydrocurcumin) that were designed for improved curcumin absorption, reducing its hydrophobicity and leading to increased membrane permeability [19,35]. For better curcumin mobility, molecules were combined with nanoparticles, micelles (e.g., liposomes) attached with phospholipid complexes [24,32,33,34,35,36,37]. Interestingly, in leukemic cells, the nanoparticles were able to increase the half-life of curcumin, as well as inhibit cell proliferation and induce apoptosis. Moreover, it was suggested that special attachment of an adjuvant to the curcumin structure can usually increase its activity [38].

Curcumin has a potential to modulate the activity of transcription factors and affect gene expression potential, as well as transcription factors (e.g., NF-κB, AP-1, STAT, ATF3, Nrf2), kinases (mitogen-activated protein kinases (MAPK), Janus kinase (JAK), hypoxia-inducible factor (HIF-1), mammalian target of rapamycin (mTOR), Ca^2+^-dependent protein kinase (CDPK), protein C phosphokinase (PKC), phosphokinase A (PKA), phosphokinase B/AKT), growth factors (epidermal growth factor receptor (EGFR), transforming growth factor (TGF-α), vascular endothelial growth factor (VEGF), fibroblasts growth factor (FGF)) and apoptosis-related proteins to indicate its anti-cancer activity [39]. Curcumin levels pleiotropic function and can modulate activity of DNA polymerase, protein kinase [40], lipoxygenase [41], or tubulin [42]. Additionally, curcumin, similarly to quercetin, can bind metal ions (Cu^2+^, Ca^2+^, Mn^2+^ or Zn^2+^) [43,44].

One of the molecular targets for curcumin seems to be NF-κB (Figure 2). It was reported that curcumin may inhibit cell growth and proliferation in breast, colorectal [45,46,47,48], lung [49] and prostate cancer [50,51]. In melanoma cell lines incubated with curcumin extract, an inhibition of cell proliferation and increased level of apoptosis was associated with downregulation of NF-κB activity [52,53].

Another transcription factor, STAT3 (Figure 2 and Figure 3), is overexpressed in many cancers. It was reported that curcumin might block the activation of STAT3 at Tyr705 residue and downregulate gene expression, thus promoting cell inhibition of proliferation and apoptosis induction in pancreatic cancer cells [54]. The administration of curcumin to a mouse with lung cancer and colon carcinoma reduced STAT3 phosphorylation at Tyr705 and caused inhibition of cyclin D1 [55]. Curcumin can affect cell signaling related to cell proliferation (cyclin D1, c-myc), as well as signaling related to cell survival (antiapoptotic, e.g., Bcl-2, Bcl-x, cIAP1, or proapoptotic (caspase-3, -9 or -8), cell cycle regulators (P53, P21), or related to protein kinases (Akt, JNK, MAPK) (Figure 2, Figure 3 and Figure 4)). Curcumin can also affect factors related to inflammation (e.g., COX-2). Moreover, it can also be involved in inhibition of p65 or MMP-2, factors important in cancer invasion (Figure 3) [51]. Curcumin was previously reported to reduce the level of Bcl-2 and PCNA, both proteins usually overexpressed in cancer. Moreover, it was found that curcumin could be effective in the inhibition of Akt/mTOR signaling in vitro in prostate adenocarcinoma cell line and as prevention of prostate cancer [56]. Curcumin and some other drugs can activate TP53 and MYC-associated factors in hepatocarcinoma cells and placenta pluripotent embryonic carcinoma cells. Curcumin was also able to inhibit NOTCH1 activity in hematopoietic Raji cells. Additionally, it is possible that, generally, we can have factors such as transformation/transcription domain-associated protein (TRRAP) that cross-links to c-Myc and is able to regulate transcription. Curcumin can mediate the covalent binding of MYC to TRRAP, and this complex leads to a growth arrest in the concentration of curcumin (30–40 µM) [57,58,59]. This confirms that gene expression can be related to many different factors. It can also be dependent on the constituents of our diet.

Cancer development and progression is associated with the disruption of the epigenome. There are some studies showing that curcumin might affect cancer cells by the modulation of epigenetic modifications [60]. Curcumin has the potential to decrease global hypomethylation in leukemic cells [61], ovarian cancer [62], melanoma [63] and breast cancer cell line in nanomolar concentrations. Curcumin blocks the catalytic center of DNMT1 and decreases the expression of demethylase at mRNA and protein levels [64,65,66]. In the mice model, it was demonstrated that curcumin reversed the methylation status of the first five CpGs in the promoter region of *Nrf 2* [67]. In the human prostate cancer model LNCaP, curcumin was able to induce the demethylation of the first 14 CpGs regions in the DNA promoter site of *Neurog1* [68]. Additionally, it was indicated that in cellular models, curcumin reduced histone acetyltransferase (HAT) activities, as well as being able to increase the level of histone acetylation and downregulate HDAC expression [69]. Moreover, curcumin should be consumed by smokers because it can help reverse the effect induced on HAT activity by cigarette smoke [70]. Curcumin efficiently blocks hyperacetylation induced by HDAC inhibitor in the prostate cancer PC3M cell line [71]. EP300/CBP inhibition by curcumin leads to the suppression of histone acetylation and expression. HAT activity decreases in hepatocarcinoma cells after curcumin exposure [72]. Curcumin administration in Raji cells decreases the level of HDAC1, HDAC3 and HDAC8 [73]. 

Curcumin can also affect miRNA expression. In breast and leukemic cancer cells, curcumin administration inhibits cell proliferation and apoptosis induction via upregulation of miR-15a and miR16 (Figure 3). Both micro RNAs are suppressors for the expression of genes related to leukemogenesis–*Bcl-2* and *WT1* (Wilms’ tumor gene 1) [74,75]. 

Curcumin’s anti-cancer activity is often associated with induction of apoptosis, autophagy and cell cycle arrest [76]. It has been demonstrated that curcumin displays the potential to sensitize colorectal cancer cells to chemotherapy. Curcumin can prevent apoptosis of tumor T cell through Stat-5a-mediated Bcl-2 induction [77]. The increased levels of Bax, Bak, Bim, Bid and Apaf-1 usually coincide with the release of cytochrome c from the outer mitochondrial membrane, generating ROS (reactive oxygen species) formation and influencing mitochondrial metabolism. Elevated ROS can usually decrease mitochondrial membrane potential [78]. 

Another anti-cancer property of curcumin seems to be overcoming multidrug resistance [79]. It was reported by Choi at al. that curcumin inhibits P-glycoprotein expression in mouse leukemia L1210/Adr cell lines. Cells treated with a combination of PI3K inhibitor and curcumin displayed lower P-glycoprotein expression in comparison to cells treated only with PI3K inhibitor. It was suggested that curcumin may act by inhibition of the PI3K/Akt/ NF-κB pathway (Figure 3). Curcumin might also block the expression of NF-κB-mediated gene *MDR1B* [80].

Moreover, curcumin is able to induce apoptosis in stem cells and proteolytic activation of caspase-3, -8 and PARP cleavage. Interestingly, the diversities in the expression of Bcl-2 family proteins were also noticed in cells [76,78]. 

Curcumin was found to be a safe and non-toxic dietary supplement. The efficacy of curcumin was studied during clinical trials [81,82]. 

In the mouse model, a high curcumin supplementation (100 mg/kg) was able to improve glucose and insulin intolerance through activating the AMPK pathway, showing the potential involvement of curcumin in metabolism [83]. Moreover, it must be underlined that curcumin can also protect against Alzheimer disease [84]. 

## 3. Acetogenins from *Annona muricata*

*Annona muricata*, commonly known as graviola (soursop, guanabana, paw-paw, sirsak), is a natural fruit tree belonging to the plant family Annonaceae. More than 400 compounds of acetogenins were described. This tropical plant grows in India, South and Central America, Indonesia and the Philippines [85,86,87]. Previously, graviola leaves were used in traditional folk medicine to treat several diseases, i.e., infections or inflammations, as well as insomnia, diabetes, headaches, arthritis, malaria or parasites [88,89]. Among the chemical compounds found in graviola extract, the anti-cancer agent seems to be acetogenin (Figure 1F) [85]. Graviola is usually presented in a leaf-powder form and can influence free radical production and the level of ROS by scavenging peroxy and nitrogen radicals [90]. Experimental data revealed that this natural extract can also protect against ROS formation by the upregulation of antioxidant gene expression, i.e., superoxide dismutase 1 (SOD1) or nuclear erythroid 2-related factor 2 (Nrf2) (Figure 3), having a potential protective function in preventing several diseases related to free radical formation, including cancerogenesis. Moreover, high levels of flavonoids or phenolic acids have been found in graviola leaves, as well as other constituents, such as rutin and vitamins. These could serve as graviola’s bioactive compounds and potent antioxidants with protective function [89,91].

Studies carried out on ethyl acetate extract of graviola leaves revealed that phytochemical compounds included in the extract were active on colon cancer cell lines HT-29 and HCT-116 and showed a cytotoxic effect toward studied cells, arresting cancer cells at the G1 phase of cell cycle and leading to the induction of apoptosis [92]. The effect of graviola leaf on stem cell extracts was also studied. Obtained results revealed that graviola can inhibit cell growth and motility. Graviola was able to promote cell cycle arrest via p21, leading to cell cycle blockade between G_0_/G_1_ phase (Figure 4), with a reduced expression of cyclin A/CDK2 and cyclin B/CDK 1 in a dose-dependent manner. 

It was also documented that graviola might display activity on cancer cell metabolism, as well as cancer formation and metastasis. It can induce changes in protein expression, leading to the downregulation of genes (*HIF-1a*, *NF-κB*, *GLUT1*, *GLUT4*, *HKII*, and *LDHA*) it can be involved in hypoxia or glycolysis of pancreatic cells. Graviola can inhibit cellular metabolism, leading to necrosis. The results of in vitro studies also confirm that graviola extract inhibits multiple signaling pathways. The results obtained during studies revealed that graviola extract inhibits tumorigenic properties of pancreatic cells and cell signaling [89]. Therefore, graviola offers a novel and promising naturally derived phytochemical that inhibits tumorigenicity and metastasis of pancreatic cancer cells in vitro and in vivo by altering cell metabolism. Such influence could also be a reason for cell cycle arrest. Moreover, this natural graviola extract, in high doses, is able to force pancreatic cells into necrosis that might induce inflammation. The diversities in cell signal transduction caused by graviola may impact on cell cycle arrest and force cancer cells to die by apoptosis or necrosis [88,89,92,93,94,95].

The graviola extract that was used for experiments with neurons induced neurotoxicity [94,96]. Moreover, the graviola extract caused cell death of mesencephalic neurons at the concentration of 1 µg/mL. Lannuzel et al. reported that annonacin from graviola extract disturbs energy production in neuronal cells, and this could be the main reason why graviola induces an apoptosis or necrosis in neurons, depending on the dose. Extracts were toxic for dopaminergic neurons, even in low concentration 0.018 µM [95,97]. These studies conducted with graviola extract on *C. elegans* mutants showed that exposure of graviola (5 mg/mL) to *C. elegans* led to the reduction of fly locomotion and its reproductive functions [96,98]. It is postulated that the studies on graviola extracts should, in future, focus on optimization of graviola extract concentration to increase its anti-cancer activity and minimize side effects. 

## 4. Resveratrol

Resveratrol [3,4′,5-trihydroxystilbene] (Figure 1B) belongs to a group of polyphenols called stilbenes [99]. Stilbenes contain two phenyl rings bound with an ethene bond. Resveratrol has three hydroxyl groups, two inside ring A and one inside ring B, and it is well soluble in fats. It is well known that stilbenes are produced by plants, usually in stress conditions, when they are damaged or exposed to UV radiation or fungal infections. Resveratrol can present in many forms. In plants, it might bind with glucose. In fruits, it presents as a glycoside, while in red wine, resveratrol occurs as an aglycone. 

In the twentieth century, it was suggested that resveratrol from red wine might protect against cardiovascular diseases. French people drink more red wine than people in other countries. Investigators combined regular red wine consumption with a decrease in cardiovascular problems and described it as the French Paradox. Unfortunately, the bioavailability of resveratrol is very low. This compound may reach only about 2 μmmol/dm^3^ in plasma, which is not enough to obtain anti-cancer protection [100].

Resveratrol is commonly known as an antioxidant agent. It has recently been reported that resveratrol may display anti-cancer properties. In a mouse skin cancer model, resveratrol inhibited tumor growth by p65 and NF-κB inhibition (Figure 3) [100]. The anti-cancer properties of resveratrol were also tested in a rat liver carcinogenesis model. It was reported that this agent can strongly suppress tumor growth by p65 attenuation, from cytosol to nucleus, and I-κB stabilization [101]. Moreover, this compound revealed the positive effect on prostate cancer by the inhibition of HIF-1α-mediated androgen receptor signaling and reducing proliferation of prostate cancer in vitro and in vivo (Figure 2 and Figure 3) [102,103,104]. Resveratrol was also able to decrease VEGF expression through the disruption of HIF-1α expression-coincided rat liver carcinogenesis [105]. The anti-cancer properties of resveratrol were shown in gastric cancer. It was associated with the inhibition of the Hedgehog signaling pathway. In gastric cancer, neoplastic cell invasion, epithelial–mesenchymal transition or metastasis were reduced after resveratrol administration [105]. Resveratrol displayed an antioxidant effect in normal cells, gaining the opposite function in cancer cells. The final activity of resveratrol could also be associated with specific microenvironmental conditions or the biology of cancer cells. It was proposed that cancer stem cells may be a good target for resveratrol because of the antioxidative potential for DNA damage and cell death in hypoxic condition [105].

It has been documented that cancer stem cells (CSCs) are the main source of tumor renewal or drug resistance. Therefore, affecting CSCs remains a successful way to overcome such complications during a patient’s treatment and outcome. The results of studies revealed that resveratrol induced apoptosis in pancreatic cancer stem cells at 10–30 μM concentrations. The same experiment also indicated that after resveratrol administration, the expression of CSC maintaining factors, such as Nanog and Oct4, was lower, as was the level of antiapoptotic protein Bcl-2 (Figure 3). Moreover, the decrease in the level of EMT proteins, Snail and Slug, was also noticed. The expression of pump efflux protein ABCG2 was also reduced after treating CSCs with resveratrol [104,106].

Resveratrol inhibited pancreatic cancer stem cells in humans and an expression of K-ras in transgenic mice by the blockade of epithelial–mesenchymal transition [107].

Moreover, resveratrol was able to decrease the level of miRs (miR-17, miR-21, miR-25, miR-92a-2, miR-103-1,-2), usually characterized as OncomiRs, in human colon cancer [108]. In prostate cancer [109] and melanoma, resveratrol decreased miR-221 levels [110], which could prove that increased level of resveratrol in the diet can have a curative role. It also confirms that diet may be important as an anti-cancer prophylactic. 

Additionally, the level of miR-542-3p and miR122-5p could be indicative of estrogen-relative or non-estrogen-relative triple-negative breast cancer cells. A decrease in miR-542-3p and an increase miR-122-5p was observed in estrogen-responsive, triple-negative breast cancer cells. When only the level of mir-122-5p increased, this was connected with triple-negative breast cancer epitope but not related with estrogen activity [111]. 

Resveratrol can induce epigenetic modifications. For example, resveratrol might downregulate the expression of metastasis-associated protein 1 (MTA 1), a part of nucleosome remodeling complex (NuRD) that leads to histone deacetylation. This complex was overexpressed in several tumor models. Resveratrol disrupts MTA1/HDAC complex interactions, leading to an increase in the expression and acetylation of P53. P53 can regulate Bax expression and leads to apoptosis. Moreover, resveratrol also strongly promotes acetylation of PTEN, which leads to the AKT pathway inhibition in prostate cancer [112]. MTA1 is a negative regulator for PTEN activity. Moreover, inhibition of tumor suppressor PTEN should usually promote AKT signalization, which is usually overactive in cancer. Thus, resveratrol induces protein acetylation and interactions with AKT, modified by phosphoryl group, leading to signal transduction inhibition. Resveratrol, in particular circumstances, is also able to activate SIRT1 and induce autophagy [112]. Interestingly, in human pancreatic and prostate cancer cells, resveratrol can regulate global gene expression by deacetylation of FOXO transcription factor [113,114]. The experimental data suggest that the downregulation of FOXO1 correlated with cell survival of cancer cells, leading to the conclusion that this protein is important as a regulator for genes that are important for cell cycle arrest [115]. 

Resveratrol can regulate the level of miRNA, thus leading to changes in gene expression. In colorectal carcinoma, resveratrol downregulated oncogenic miRNAs (miR-21, miR-25), usually overexpressed in this cancer [116]. Downregulation of miR-21, miR-30a-5p and miR-19 expressions by resveratrol inhibited proliferation of glioma cells, promoting cell cycle arrest or switching to apoptosis [117]. Downregulation of miRNAs led to decreased expression of PTEN, EGFR, STAT3, COX2, NFkB and PI3/AKT/mTOR, which are described as pivotal proteins, important for glioma development [117,118]. 

Resveratrol was reported as a safe dietary supplement. However, since it normally depends on the dose, 1.0 g of resveratrol appeared to be non-toxic. During long-term clinical trials, adverse effects in the used doses of resveratrol were not observed [119,120,121,122,123,124]. Single and repeated doses of up to 5 mg/mL were well tolerated by healthy volunteers. Additional studies should be performed to establish safe resveratrol doses for other diseases. 

Interestingly, some side effects (such as nausea, diarrhea, vomiting and liver dysfunction) were observed in patients with non-alcoholic fatty liver disease, which may be correlated with liver problems and digestion. Moreover, rabbits orally supplemented with resveratrol unfortunately had significantly more atherosclerotic changes on aortic surface than non-treated rabbits, which suggests that rabbits are not a good model for human atherosclerosis studies [125]. Some nephrotoxicity caused by resveratrol was also noticed in another study. Rats were orally supplemented with resveratrol for 28 days. Urea nitrogen and creatinine levels were elevated in the blood serum. Additionally, some histological changes were detected in the kidneys [126]. The understanding and dose dependency of resveratrol were complicated by the fact that orally administrated resveratrol is digested by gut microbiota. It is possible that the observed side effects were a reason for disturbances in the liver function caused by fat and additional disturbances caused by resveratrol [124]. Interestingly, when we did combine the above studies related to rabbits or rats, we concluded which animals are sometimes not good models for studies on atherosclerosis, similarly to the case of pig sera, where, during experimental lab lessons for clinical biochemistry, it was difficult to determine the level of cholesterol. 

## 5. Epigallocatechin-3-gallate

Epigallocatechin-3-gallate (EGCG, Figure 1D) belongs to a group of polyphenols extracted from green tea. Its anti-oxidative, anti-inflammatory and anti-cancer properties were noticed in many experiments. While the accurate mechanism of EGCG action is not fully understood, the results of studies reveal that this agent could be a very promising chemopreventive compound in anti-cancer therapy [127]. 

It has been reported that EGCG seems to be active in the suppression of the NF-κB pathway, leading to cell growth inhibition and apoptosis (Figure 2 and Figure 3) [128]. In human colorectal cancer cell lines (HT-29 and HCA-7), EGCG led to COX-2-promoter inhibition through downregulation of NF-κB [129]. In human bronchial epithelial cells, EGCG inhibited NF-κB activity caused by cigarette smoking [130]. The administration of green tea polyphenols to a mouse with human melanoma and patients with breast cancer sensitized cancer cells to radio- and chemotherapy by the downregulation of the NF-κB pathway [131,132]. EGCG, in combination with curcumin, may disrupt STAT3 signaling in breast cancer stem cells [133]. Additionally, green tea polyphenols may potentiate gemcitabine activation and promote PARP cleavage and caspase-3-related apoptosis in pancreatic cancer cells, as a consequence of STAT3 inhibition [134]. Experimental data revealed that EGCG strongly binds to Arg609 in the SH2 domain of STAT3. Such interaction leads to the inhibition of STAT3 phosphorylation at Tyr705, as well as to downregulation of the STAT3 pathway [133]. Moreover, EGCG was able to block tumor cell proliferation in breast, bronchial and stomach cancer cells by the inhibition of AP-1 activity [128,135]. EGCG was also more effective for different types of treatment [136,137]. 

Recent investigations revealed that polyphenols from green tea may exhibit anti-cancer properties as a result of changes in epigenetic machinery. It was suggested that EGCG may act as a DNA methyltransferase DNMT1 inhibitor, thus being able to regulate tumor suppressor genes that are hypermethylated in many types of cancer [138,139]. Molecular modeling studies revealed that B- and D-rings play a role in inhibitory effect on DNMT activity. Interestingly, EGCG was able to interact with Pro-1223, Glu-1265, Cys-1225, Ser-1229 and Arg-1309 in the DNMT catalytic center [140,141]. It was observed that the *GSTP1 gene* from prostate cancer was reactivated after demethylation caused by green tea polyphenols (Figure 3). The methylation level of catechol groups in EGCG was probably the reason for a decrease in targeted gene methylation status, i.e., *GSTP1* [142]. It is possible that methyl groups are translocated from *GSTP1* gene into EGCG residues. The depletion of S-adenosyl-L-methionine:hydroxide adenosyltransferase (SAM) may block the activity of DNMTs. DNMT activity was also modulated in colon and prostate cancer cells by EGCG. Promoter DNA regions for annexin A1 and Wnt inhibitory factor 1 were demethylated by EGCG in lung cancer cells [139,143,144,145]. 

In an epithelial tumor model, EGCG downregulated HATs expression and the acetylation level of histone and non-histone proteins, such as NF-κB. [146].

EGCG was able to decrease IDO (indoleamine 2,3-dioxygenase) level. Overexpression of IDO may lead to the immune system’s tolerance of tumor cells. IDO is responsible for converting tryptophan to kynurenine. Overexpression of this enzyme can suppress T cell action. It is associated with kynurenine generation and depletion of tryptophan in the microenvironment. Kynurenine may interact with aryl hydrocarbon receptor and inhibit T cell and NK cell proliferation. On the other hand, a rapid depletion of tryptophan from the microenvironment can induce stress signals to T cells, promoting CD4 differentiation toward regulatory T cells (Tregs) [147,148,149]. Moreover, in oral cancer cell lines, EGCG displayed the ability to decrease IDO expression by inhibiting INFγ activity [150,151]. Interestingly, green tea polyphenols can also inhibit phosphorylation of STAT1 and its translocation to nucleus. Additionally, EGCG might decrease expression and the level of phosphorylated PKC-δ and JAKs associated with STAT1 activity (Figure 3) [152,153].

Interestingly, EGCG treatment leads to phosphorylation of AMP-activated protein kinase reduction of mitochondrial membrane potential, thus inducing changes in metabolism toward the activation of lipophagy. It can be helpful in reduction of lipid contents and can prevent obesity or obesity-related metabolic disorders [154].

Studies concerning pharmacokinetic and safety profile of EGCG were conducted. Thirty-three CLL patients were orally treated with EGCG at a dose of 400–1200 mg twice a day for 4 weeks. Two of them reported side effects: dysphagia and intestinal problems. Generally, all human studies with EGCG revealed that 8–16 cups of green tea were safe and well tolerated [155]. These studies confirm a personal sensitivity to any anti-cancer agents and the necessity to choose a proper dose of anti-cancer agent(s) for some patients.

## 6. Quercetin

Quercetin belongs to a group of flavonoids (Figure 1E) [156]. This compound was found in many natural products, such as fruits (apples, oranges), vegetables (onion, broccoli, parsley) and herbs (sage), but also in tea and red wine [139]. In a diet, quercetin mostly occurs in the form of β-glycosides, usually connected with glucose or rutin. This compound displays a broad spectrum of anti-allergic, anti-viral, anti-cancer, antioxidant and immunology-related anti-inflammatory properties, as well as attenuating lipid peroxidation, platelet aggregation, anti- atherosclerosis and inducing apoptosis. Quercetin can be active in obesity [156,157,158,159,160,161,162].

It has been reported that quercetin is able to scavenge oxygen radicals, as well as inhibit enzymes involved in ROS formation, e.g., xanthine oxidase [103]. The antioxidative properties of quercetin also include the formation of inert complexes by a chelation of several metal ions [163,164], for example, cadmium [165]. Quercetin can scavenge radicals produced by cadmium, showing antioxidative and anti-cancer-activity protection. Moreover, quercetin has the ability to modulate the activity of enzymes involved in antioxidant function, such as superoxide dismutase (SOD) or glutathione S-transferase (GST) (Figure 3) [166]. Therefore, quercetin interacting with metal ions, such as nickel, can protect from downregulation of DNA repair pathways in lung cells, which might protect them from cancer [164]. The similar protective function by the chelation of other metal ions included ions characterized as food ingredients, including food preservatives or additives. 

Considering its high antioxidant properties, quercetin seems to be a promising compound for preventing carcinogenesis. High doses of quercetin were able to induce apoptosis in cancer cells and inhibit or decrease cancer progression. This could be related to the high toxicity of quercetin to cancer cells in comparison to its limited effect on normal cells. It has been reported that in higher concentration, quercetin enables apoptosis activation by the upregulation of proapoptotic protein Bax and the downregulation of antiapoptotic Mcl-1. In lower concentration, it can protect neurons from apoptosis (Figure 2 and Figure 3). Quercetin was also able to induce caspase-dependent apoptosis through an inhibition of activator of transcription and HER2 overexpression in BT-474 breast cancer cells [167,168,169,170]. Additionally, other data suggest that quercetin can also inhibit PI3K/AKT, Wnt/β-catenin and STAT3 pathways (Figure 2) [159].

Quercetin also has the ability to affect the epigenome. Its modulative effect on histone acetylation was observed in breast cancer cells, where quercetin decreased the level of cyclooxygenase-2. It might be caused by the blocking of transcription factors binding NF-kB or p300 [171,172] (Figure 4). Quercetin was found to activate histone deacetylase SIRT1 in yeast. Quercetin also inhibited expression of IP-10 (TNF-induced interferon-gamma-inducible protein 10) and MIP-2 (macrophage inflammatory protein 2) through the inhibition of CBP/p300 and an acetylation of histone H3 on the promoter regions of these genes [172]. Quercetin was able to acetylate histone H3, promoting FASL-mediated apoptosis in acute myeloid leukemia cell line (HL-60) [173]. Quercetin inhibits TNF-α induced HUVECs apoptosis and inflammation via downregulation of NF-kB and AP-1 signaling pathway in vitro. This is why, in particular conditions, quercetin can inhibit apoptosis and also inflammation [174].

Interestingly, quercetin activity is dependent on concentration. At low concentration, this agent can stimulate the proliferation of cells, showing its activity as a potential drug for neurodegenerative diseases. In a high concentration, quercetin induces apoptosis, while in the lowest concentration it can stimulate the immunological system and inhibit inflammation [174]. Recently, it was reported that quercetin can also be involved in cellular metabolism. It was suggested that quercetin displays anti-diabetic properties, acting on glucose, cholesterol or triglycerides levels [175,176]. Quercetin’s protective role in human health was also confirmed in studies on cancer cells. Quercetin can induce cell cycle G1 arrest because of increased level of Cdk inhibitors (p21, p27) in human hepatoma cell line (HepG2) [177]. Moreover, quercetin also abolishes oral cancer cell migration and invasion by the inhibition of matrix metalloproteinase-2/-9 signaling pathways [178].

Quercetin was able to inhibit the proliferation and invasion of cancer cells by upregulating miR-146a expression in breast and gastric cancer cells [179,180]. Moreover, similarly to LPS, quercetin and curcumin can block toll-like receptor 4 (TLR4) function and inhibit signalization to induce expression of nuclear factor kappa-light-chain-enhancer and therefore inhibit inflammation. This can be a common activity for molecules with anti-inflammatory function. It is not certain whether, in stressful conditions, quercetin could suppress expression of heat shock protein HSP70, or whether quercetin, because of several activities, would be able to overcome stress [181]. Interestingly, similarly to quercetin, nanoparticles such as PM2.5 upregulate microRNA-146a-3p expression and induce macrophage M1 polarization in RAW264.7 cells by reducing the Sirtuin expression, showing a potential involvement in epigenetic changes [182].

The available data suggest that the administration of quercetin does not cause serious side effects. It must be stated that, because of diverse quercetin functions under different doses, determining an appropriate dose and method of quercetin administration is necessary.

## 7. Lycopene

Lycopene is the most common carotenoid found in red vegetables and fruits [183,184]. The group of carotenoids comprises carrots, tomatoes, watermelons, guavas, papayas and grapes. It is a polyunsaturated isomer of β-carotene, containing 11 conjugated and 2 non-conjugated bonds (Figure 1C). Because of its chemical structure, lycopene was reported as a strong antioxidative agent. It has been postulated that chronic oxidative stress could be a reason for a coincidence with cancerogenesis. Therefore, a diet rich in lycopene may protect against oxidative stress, DNA damage and lipid peroxidation [183]. Another in vivo study confirmed that tomato sauce, rich in lycopene, successfully blocked mitochondrial DNA damage caused by ROS generated by UV radiation [184]. A very promising experiment in vivo on gastric cancer cells revealed that the administration of lycopene (2,5 mg/kg) was able to reduce lipid peroxidation, similarly to vitamins C and E [185].

Antioxidant response elements (ARE) are cis-DNA sequences, which are responsible for the upregulation of enzyme genes with antioxidative properties and those involved in phase II detoxification. The main ARE transcription factor is Nrf2. Nrf2 plays an important role in detoxification and modulation of oxidant defense system by the upregulation of stress-induced cytoprotective enzymes (Figure 3). It was observed that in breast and liver cancer cell lines, lycopene promoted the upregulation of genes for NAD(P)H:quinone oxidoreductase 1 (NQO1), glutamate cysteine ligase (GCL) and glutathione (GSH), by activating Nrf2 [186,187]. The mechanism of lycopene-related translocation of Nrf2 from cytoplasm to nucleus is still unknown. One hypothesis suggests that lycopene metabolites may facilitate the dissociation of Nrf2 from the complex with Keap1 protein. The release of Nrf2 from this inhibitory complex leads to translocation to nucleus, where Nrf2 may act as an active transcription factor (Figure 3) [188,189].

Growth factors, such as IGF, VEGF or EGF, promote many signaling pathways related to proliferation, differentiation and cell survival. Some experiments revealed an inhibitory effect of lycopene on signaling pathways associated with these growth factors. In breast and lung cancer cell lines, lycopene was able to reduce the level of IGF-1 and increase the level of IGF binding proteins (IGFBPs), which could sequestrate IGF. Therefore, lycopene downregulated signaling pathways dependent on IGF [190]. Similar results were obtained in a high-risk population of colorectal cancer patients. This may suggest that supplementation of lycopene may protect against colon cancer [191]. Moreover, tomato and lycopene downregulated androgen metabolism and signaling in colon cancer [192]. Another observation demonstrated that diet rich in lycopene can prevent lung cancer in the group of tobacco-smoking people. An increased level of IGFBP3 and decreased level of phosphorylated BAD were noticed. The obtained results suggest that IGFBP3 might inhibit PI3K/AKT/PKB and Ras/Raf/MAP signaling pathways and can promote apoptosis in lung cancer. Recent in vitro and in vivo investigations on prostate cancer models confirmed that lycopene may increase antineoplastic effect of docetaxel by the inhibition of IGFR1 and upregulation of IGFBP3 [193].

Cell cycle disruptions often occur during cancerogenesis. Uncontrolled division of cancer cells leads to tumor growth and progression. Therefore, cell cycle arrest is one of the most promising targets for cancer therapy. It has been postulated that in cancer cell lines, lycopene can affect proteins related to cell cycle activity. Experiments conducted on the human hepatoma Hep3B cell line revealed that lycopene led to G0/G1 cell cycle arrest [194]. In MCF-7 breast cancer cell lines, lycopene was able to downregulate cyclin D and c-myc expression (Figure 4). Cell cycle inhibition was associated with a decrease in the level of CDK4 expression, prolonged retention of p27 in the cytoplasm and cyclin E/ CDK2 complex inhibition. As a result, a decrease in the phosphorylation level of pRb and inhibition on G1/S transition were observed [195]. Additionally, in breast cancer cells, cell cycle arrest mediated by IGF1 was linked with reduced phosphorylation of tyrosine residues in insulin receptor substrate 1 (IRS1) [196]. The anti-cancer activity of lycopene via cell cycle inhibition was also noticed in LNCaP prostate carcinoma cells. Lycopene led to cell cycle arrest in the G1/S phase. It coincided with the decrease in the level of cyclin D and an increase in expression of p21, p27 and p53 (Figure 4). The mechanism of action included the inactivation of Ras [195] by the decrease in expression of 3-hydroxyl-3-methylglutarylcoenzyme A reductase (HMG-CoA). The reduction in Ras farnesylation caused decreased activity of this oncogene. Simultaneously, the reduction in expression of NF-κB-dependent proteins, e.g., Bcl-2, Bcl-X_L_ cyclin D and c-XIAP, was also observed [197].

It has been reported that programed cell death could also be modulated by lycopene [198,199,200,201]. It was demonstrated that in the HCT-29 human colon cancer cell line, lycopene induced apoptosis via death receptors. The inactivation of Akt and mTOR kinases enhanced the accumulation of Fas ligand and Bax [198]. In human gastric cancer cells, in vivo lycopene increased the level of proapoptotic proteins Bim and Bax, promoting cleavage of caspases -3 and -8, and it was also able to decrease the level of Bcl-2 (Figure 3) [201]. Other results suggest that lycopene may also mediate apoptosis induction via intrinsic pathway. In LNCaP prostate carcinoma cells, lycopene induced apoptosis by reducing mitochondrial membrane potential and led to the release of cytochrome c to cytosol [202]. Moreover, the combination of lycopene with genistein successfully led to apoptosis in chemically induced breast cancer model in vivo. It was associated with a decrease in Bcl-2 level, an increase in Bax expression and the activation of caspases -3 and -9 [203]. Lycopene also displayed synergistic activity with docetaxel in LNCaP cell line by inducing p53 expression and decreasing the level of survivin, both in vitro and in vivo [202].

Metastasis and angiogenesis are important features during cancer development and progression. Among natural compounds, lycopene exhibits very strong properties against both processes. It was demonstrated that lycopene may inhibit the expression of proinflammatory interleukin 8 (IL-8) by blocking NF-κB transcription factor (Figure 2). The reduction in ERK1/2, JNK and p38 MAPK phosphorylation, as well as increased expression of PPARγ, PTEN activity and inactivation of AKT, were also observed [204,205,206].

Lycopene in vivo was able to decrease the level of proliferating cell nuclear antigen (PCNA) and β-catenin, while an increase in p21 and E-cadherin adhesion molecule expression in colon cancer were noticed. Moreover, lycopene also decreased the level of MMP7 (Figure 3). It was reported that a decrease in the expression of MMP7 correlated with an increased level of adhesion molecule E-cadherin expression, which may suggest that MMP7 could be responsible for this cadherins cleavage. Lycopene suppressed MMP7 activity [207,208].

Other results have documented that lycopene inhibited migration and invasion of hepatoma cancer cells in vitro by the upregulation of *Nm-23H1* gene expression, which led to the suppression of metastasis [209]. The results obtained in other studies confirmed that lycopene may inhibit metastasis, migration and invasion by the downregulation of MMP2 and MMP9 [210].

Chemotherapy, radiation and surgery are the three most common methods of treating cancer, but they can still be ineffective. The biggest disadvantages of anti-cancer therapies are the side effects and resistance to treatment. There are encouraging data showing that lycopene may be used as an adjuvant in cancer therapy. The obtained data confirm that lycopene has the ability to revert multidrug resistance and induce apoptosis in tumor cells [211]. Moreover, lycopene may be used alone or in combination with other anti-neoplastic agents to fight against metastasis and cell migration, as well as to alleviate the side effects of treatment with chemically synthesized drugs [201,203,204]. Recently published studies also confirm the important role of lycopene as an important factor in decreasing the level of cholesterol, by targeting factors important in cholesterol synthesis in the liver [212].

Lycopene belongs to a group of retinoids. It is a product as important as provitamin A (carotenoid). Lycopene, similarly to vitamin D, is lipophilic, and, except for its antioxidative function, may be able to regulate gene expression. Carotenoids are necessary for proper vision, hematopoiesis, reproduction (embryogenesis), metabolism and bone structure [213]. The mechanism of lycopene function, as other retinoids, can form 9-cis-retinoic acid, a ligand for retinoid X receptors (RXR). This receptor can dimerize with vitamin D receptor (VDR) and vitamin D, as well as possibly other factors important in the regulation of gene expression [213].

In a century directed toward personalized therapy, we have to remember that designing a personalized therapy strategy for the patient increases the chance of a response to treatment and increases the patient’s lifespan because of several targets that agents usually react to [214,215,216]. Natural anti-cancer agents, consisting of chemical structures reflecting anti-cancer potency, are usually safer to use because of reduced side effects. In future, we will need to design special transport methods to carry anti-cancer molecules to cancer tissues and calculate curative doses of potential drugs to optimize their anti-cancer activity.

## 8. Clinical Trials Involving Natural Compounds

There are several clinical trials on anti-cancer activity, including on the use of natural compounds. Such anti-cancer agents are usually tested as a single agent or in combination with other agents or drugs used in anti-cancer treatment (clinicaltrials.gov).

Curcumin was tested as a potential anti-cancer agent, mainly for solid tumors, e.g., breast, pancreatic, colon, endometrial or prostate cancer. Interestingly, there was also a complete trial (NCT01160302) using curcumin as a biomarker in cancer development for newly diagnosed patients with head and neck squamous cell carcinoma. This trial seems to be of special importance because of curcumin’s potential anti-cancer function, and it could be of significant importance in cancer prophylactics. Curcumin was also studied as a chemopreventive agent for colorectal cancer (NCT00118989). Curcumin was tested as a modification to FOLFOX chemotherapy. However, there was no significant difference between groups receiving FOLFOX therapy with or without curcumin [217]. In another study, a combination of Meriva^®^ (curcumin with phospholipids) with addition of gemcitabine revealed a good response rate in the first line of treatment of advanced pancreatic cancer [218]. Curcumin combined with other natural compounds (APG-157 botanical drug) exhibited some anti-cancer potential for oral cancer. The level of cytokines related to inflammation (IL-1β, IL-6 and IL-8) dropped in the saliva of cancer patients, which might suggest their anti-inflammatory potential. Moreover, the expression of factors involved in T cell differentiation and their activation originating from tumor microenvironment was increased [219]. The anti-cancer effect of curcumin was also observed in patients with prostate cancer. Patients received 1440 mg of curcumin orally every day for 6 months. The group of tested patients demonstrated lower concentration of PSA in comparison to patients receiving a placebo [220]. Curcumin was also tested as a potent anti-cancer therapy for glioblastoma. A micellar form of curcuminoids was administrated three times daily to patients with glioblastoma 4 days prior to surgery, in corresponding doses (57.4 mg curcumin, 11.2 mg demethoxycurcumin and 1.4 mg bis-demethoxycurcumin). Curcuminoids were present in the plasma, as well as in the tumor tissue. The level of inorganic phosphate in curcuminoid-treated patients increased. Moreover, the mean intratumoral pH increased after curcuminoids administration. The authors suggested that curcuminoids in glioblastoma exhibit antitumor efficacy, disturbing tumor metabolism [221].

Graviola was tested for its anti-cancer activity, as well as its potential protective function against cancer. Studies were performed on human colorectal cancer cells (ex vivo and in vitro) [222]. For the purpose of evaluation, the effect of consecutive ingestion of graviola extract for a period of 8 weeks was studied in patients with colorectal cancer (NCT02439580). The obtained results were correlated with nutritional diet status, quality of patients’ lives and the level of inflammation.

There are also several clinical trials into the anti-cancer efficacy of resveratrol. While resveratrol was studied and finally withdrawn in the monitoring of hepatocyte function in cancer (NCT02261844), its anti-cancer activity was also analyzed in colon cancer (NCT00256334). Interestingly, there is a complete trial on resveratrol activity on Notch-1 signaling in low grade gastrointestinal tumors (NCT01476592). Resveratrol function was also tested on polycystic ovary syndrome (NCT01720459). Patients treated with resveratrol (800 mg per day) for 40 days revealed a decreased level of inflammatory-related factors, i.e., IL-6, IL-1β, IL-18, TNF-α, NF-κB and CRP protein. Additionally, the expression of ATF4 and ATF6 was increased, while the expression of CHOP, GRP78 and XBP1 dropped [223]. In other studies, patients with polycystic ovary syndrome received resveratrol at the dose of 800 mg/day for 40 days. The level of sex-related hormones, as well as VEGF and HIF1 expression, were analyzed after treatment. Patients administered with resveratrol exhibited a difference in the level of TSH, FSH, LH and testosterone. The reduced expression of VEGF and HIF1 were observed in granulosa cells. Additionally, the high-quality oocytes and high-quality embryos were obtained in the group of patients treated with resveratrol [224]. The antitumor effect of resveratrol was also detected in colorectal cancer and hepatic metastases [225,226,227,228]. Before tumor resection, patients with colorectal cancer consumed resveratrol eight times a day in doses of 0.5 and 1.0 g, respectively. During the experiment, resveratrol was well tolerated. This compound was detected in the tumor tissue, and the level of Ki-67 was decreased [227]. In another study, patients with colorectal cancer and hepatic metastases were treated with micronized resveratrol–SRT501 for 14 days at 5 g/day. Resveratrol accumulated in the metastatic sites in the liver. It was correlated with an increased level of cleaved caspase 3 [228].

Clinical trials on green tea extracts were tested as a prevention for colorectal adenomas and colorectal cancer (NCT02321969), as well as for analysis of anti-cancer activity in bladder cancer (NCT00666562), breast cancer (NCT 00949923) and prostate cancer (NCT01340599, NCT00459407, NCT00253643). The potent function of such extracts was also analyzed in prevention of cancer for healthy individuals (NCT00091325), patients with breast cancer risk (NCT00917735) and with liver cancer (NCT03278925). After the studies, a protective effect of green tea polyphenols for breast cancer was described [229]. Breast cancer patients, in the early stages of disease development, received 44.9 mg of EGCG for 4 weeks before surgery. The EGCG was detected in the tumor tissue and plasma. Anti-cancer efficacy of EGCG was observed. The level of free EGCG was positively correlated with decreased level of Ki-67 in the tumor tissue. The protective effect of green tea polyphenols was also confirmed by another study. Capsules with 843 mg of EGCG were administered to post-menopausal women with an increased risk of breast cancer. In the group of younger patients (50–55 years), such supplementation significantly reduced changes in mammographic density. It was demonstrated that intake of EGCG may reduce changes related to age in the group of patients with high risk of breast cancer. Moreover, drinking green tea could prevent breast cancer in postmenopausal women [230]. Green tea polyphenols in combination with curcumin revealed, when supplemented orally, a great potential in the reduction of malignant disorders. In these studies, patients received EGCG (800 mg/day) and curcumin (950 mg/day) for three months. The synergistic effect of natural compounds activity was observed. The clinical response rate after supplementation was determined for the combination of green tea polyphenols and curcumin (65%) in comparison to curcumin alone (11.55%) and green tea alone (7.35%), respectively. The expressions of Ki-67, P53 and cyclin D1 were downregulated in the samples collected after the experiment [231]. Anti-cancer properties of EGCG were also observed in colon carcinoma clinical trials. The potential anti-cancer effect of green tea extract was analyzed in Korean patients after resection of colorectal carcinomas. One group of patients received green tea extract (900 mg per day) for 1 year, whereas the second group received a placebo. It was demonstrated that the risk of adenomas and relapse adenomas decreased in the group of patients who received green tea extracts [232].

Quercetin was used in the prevention of metastasis in the group of colorectal (NCT00003365) and prostate cancer patients after surgery (NCT01912820). There is a complete trial studying the effect of quercetin as a prevention or treatment for oral mucositis (NCT01732393). Quercetin activity is currently analyzed as an anti-cancer compound in pancreatic cancer [233]. Moreover, a clinical trial involving broccoli sprouts is being conducted. Based on the knowledge that sulphurate, isothiocyanate and quercetin inhibit tumor growth and sensitize tumor cells to the treatment, the current study will provide more information about broccoli sprout activity in the treatment of advanced pancreatic ductal adenocarcinomas. In another study, quercetin was administrated to patients, and this compound was able to inhibit in vivo tyrosine kinase activity. Additionally, quercetin was also supplemented to patients with ovarian cancer, which is resistant to cisplatin. The level of the cancer marker CA 125 in this patient group decreased from 295 before to 55 U/mL after treatment. The other patient group with hepatoma administered with quercetin revealed a decreased level of alpha fetoprotein in the serum. Quercetin was also able to decrease an expression of tyrosine kinase, which could also be connected with antitumor activity [234].

Interestingly, lycopene, which is present in tomatoes, has anti-cancer properties that have been tested in metastatic prostate cancer (NCT00068731). Patients with prostate cancer received a lycopene-rich diet with 15 mg of lycopene, twice daily. For one patient with androgen-independent prostate cancer, lycopene was not effective [235]. Interestingly, another patient with prostate cancer who received a diet with a higher amount of lycopene (30 mg per day) seemed to be active on lycopene. For this patient group, the PSA level decreased when patients were supplemented with lycopene, or lycopene in combination with three fatty acids and selenium. After these studies, the authors suggested that the effect of response was dependent on cancer aggressiveness and the number of supplements in the blood circulation [236]. Lycopene activity, combined with green tea extract, was also tested as a supplement in a man with increased risk of prostate cancer [237,238]. After 6 months of trial duration, the changes in patient metabolome were investigated. Lycopene and polyphenols from green tea caused a decrease in the level of acetate, valine and pyruvate. The decrease in pyruvate was particularly interesting and might be correlated with a lower risk of prostate cancer. The positive activity of lycopene in the diet was also confirmed in the study of breast cancer patients who terminated chemotherapy because of weakness. Moreover, a diet containing a higher amount of lycopene was more likely to reduce fatigue in comparison to standard diets. Interestingly, a diet consisting of fruits, vegetables, omega-3 fatty acids was indicated as a non-toxic way of treatment and a strategy to improve the well-being of fatigued patients [239].

## 9. Conclusions

For anti-cancer prophylactic purpose, special anti-cancer antioxidants in proper doses should be introduced into the human diet or supplemented to optimize human body function. Therefore, a diet including lots of vegetables and fruits is healthy because of the presence, for example, of quercetin in onion and broccoli, which together with other natural substances, especially flavonoids (e.g., curcumin) can provide anti-cancer protection for the human body. This is why the constituents of our meals could become a natural anti-cancer prophylactic and will improve our health condition. It must be stated that to reach anti-cancer protection through the optimal level of glucuronic acid, we would have to eat about 2 kg of apples each day, and this is usually not possible to achieve. However, we can divide the total amount of antioxidative protection constituents in our diet into smaller portions of several other products, e.g., other fruits or vegetables rich in glucuronic acid, such as grapes, grapefruits and broccoli, which have similar antioxidative potential.

The number of anti-cancer agents and/or the antioxidative protective potential in food should be experimentally validated, and the dose of such healthy ingredients should be marked on the product. Another way to increase anti-cancer protection is to add agents with anti-cancer activity or other constituents with antioxidative potential to food and consume these during meals, in order to take these constituents in the doses necessary to be protected against cancer.

Importantly (once all the potential values have been determined), a daily amount of compounds with anti-cancer protection should be determined, standardized for prophylactic use, and all food products boxes should be marked showing the value of anti-cancer protection, i.e., the value of anti-cancer/antioxidant protection contained in the product as a percentage of the daily required amount of protection necessary for men, women and children, similar to today’s nutritional value on information guides. Such standardization will be helpful in conscious decision making during shopping. Natural agents could also be used as anti-cancer drugs. For optimization of natural compounds’ anti-cancer potency, these agents should be closed inside drug carriers and designed for directing drug transport toward cancer cells. The carriers should be translocated inside blood stream to cancer cells and be able to recognize and bind to characteristic receptors or other markers present on cancer cells. Moreover, drug-carrier molecules should be able to release anti-cancer agent(s) to cancer cells, inducing cell death, after docking to cancer cells. This way of transport should increase natural compounds’ anti-cancer activity and simultaneously reduce or avoid potential side effects.

## Figures and Tables

**Figure 1 ijms-22-13659-f001:**
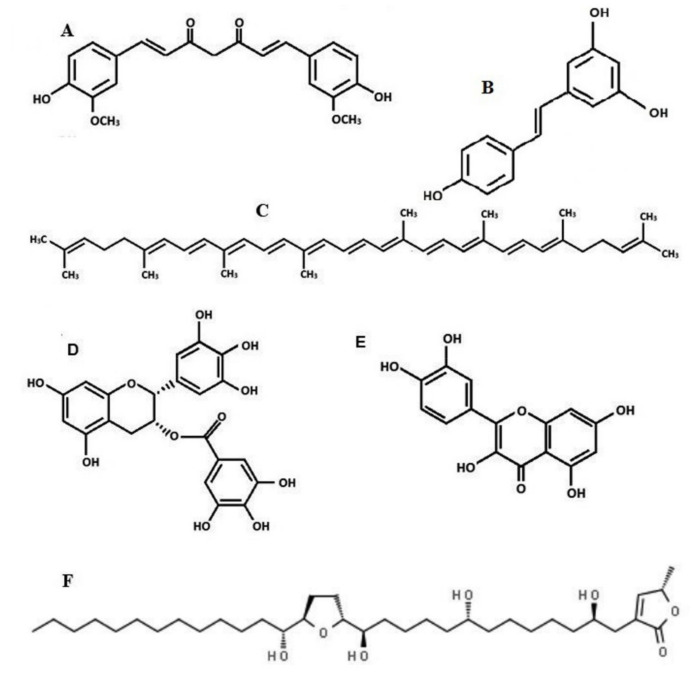
Chemical structures of natural compounds with anti-cancer potential, i.e., curcumin (**A**), resveratrol (**B**), lycopene (**C**), epigallocatechin-3-gallate (**D**), quercetin (**E**), acetogenin from graviola extract (**F**).

**Figure 2 ijms-22-13659-f002:**
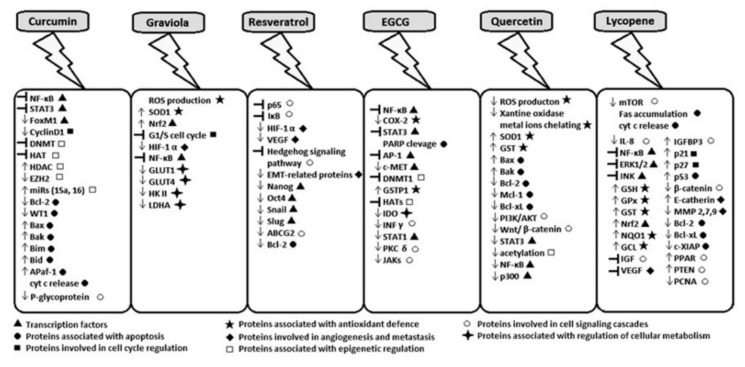
Molecular targets for natural compounds, i.e., transcription factors, proteins related to apoptosis, cell cycle, cell signaling, epigenetic regulation of gene expression, factors involved in angiogenesis, as well as in metastasis or cellular metabolism, factors related to cell antioxidant defense (differences in expression; ↑—increase, ↓—decrease, ┤—inhibition).

**Figure 3 ijms-22-13659-f003:**
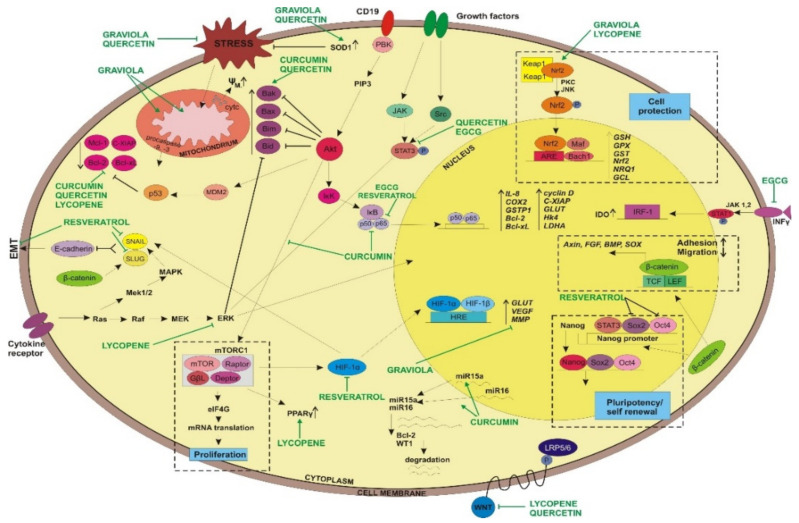
Natural agents’ molecular targets in cells involved in cell signaling, apoptosis, self-renewal, cell protection, adhesion and migration.

**Figure 4 ijms-22-13659-f004:**
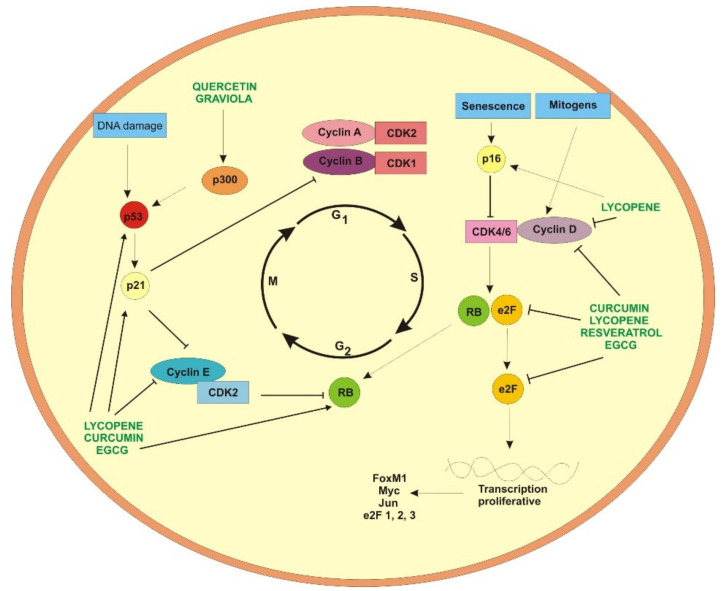
Natural compounds’ activity on factors involved in cell cycle.

## Data Availability

Not applicable.

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
