# Peer review of "Molecular Targets of Natural Compounds with Anti-Cancer Properties"

_ijms, 2021, doi:10.3390/ijms222413659_

Round 1

Reviewer 1 Report

  1. The quality of the illustrations needs to be improved. Figures are hard to read
  2. The text of the review is written carelessly. Requires careful proofreading.
  3. Chapter titles should be more descriptive
  4. To improve the quality of the manuscript, tables should be built and additional illustrations prepared.
  5. References are designed carelessly and do not correspond to the rules of the journal

Author Response

Respond to review #1:

  1. I did attach 333 dpi files, so from my point of view this is a digital quality (higher than 300 dpi). If Figures are not good, it could be because of some problems from reviewer computer side.
  2. Text was modified only in a few parts and manuscript was corrected previously by a person who work as an English language translator/corrector. Additionally, the quality of language corrections was also confirm by English language native speaker. I can provide certificate.
  3. If I include all information about for example quercetin or graviola, I do not understand why the name of natural substance is not enough to describe what will be written about in this chapter?
  4. It is possible that everybody could have ana additional ideas how to improve the quality of manuscript. I would like to publish this manuscript as soon as it is possible instead of introducing next changes that maybe will provide some information in additional table but we must understand what the most important is a time, when peoples will be able to reed manuscript, because I hope it could change and maybe make some progress in more effective cancer treatment, and what is the most important to give for scientist a new ideas to improve anticancer patients efficacy.
  5. I do not know how it was possible that I did prepared during my holidays correct version of manuscript and this version name was given in CAPITAL letters. This version I have still on my computer. Additionally, because of several problems previously and long time which I spent for references converting, therefore I did check before manuscript confirmation that the right version was upload. But currently the other file is combined with submission. I could not took and print the submission and I will not comment why this file is exchange. I add the correct version of manuscript with this rebuttal letter.

Reviewer 2 Report

This is an excellent review manuscript about a very interesting and demanding topic

For anticancer prophylactic purpose, a special anti-cancer anti-oxidants in a proper dose should be introduced into the human diet, or supplemented to optimize human body function. Therefore, a diet included lots of vegetables and fruits are healthy, because of presence for example in onion and broccoli a quercetin, that together with other natural substances, a specially flavonoids (eg, curcumin) could provide anticancer protection for human body.

That is why the constituents of our meals, could become a natural anticancer prophylactic and will improve our health condition. It must be stated that to reach anticancer protection through the optimal level of glucuronic acid we would have to eat about 2 kg of apples each day, and this is usually not possible to reach. However, we can divide the total amount of anti-oxidative protection constituents in our diet into smaller portions of several other products, e.g. other fruits or vegetables rich in glucuronic acid, such as grapes, grapefruits and broccoli, which have similar anti-oxidative potential

The number of anti-cancer agents and/or the anti-oxidative protective potential in food should be experimentally validated, and the dose of such healthy ingredients should be marked on the product. Another way to increase anti- cancer protection is to add agents with anti-cancer activity or other constituents with anti-oxidative potential to food and consume these during meals, to take these constituents in the necessary doses to be protected against cancers

Importantly, (once all the potential values have been determined) a daily amount of compounds with anti- cancer protection should be determined, standardized for prophylactic use, and all food products boxes should be marked showing value of anti-cancer protection, i.e. a value of anti-cancer/anti-oxidant protection contained in the product as a percentage of the daily required amount of protection, necessary for men, women and children, similar to today’s nutritional value of information guides. Such standardization will be helpful in conscious decisions-making during shopping.

For optimization of natural compounds anti- cancer potency, these agents should be closed inside drug carriers, designed for directing drug transport toward cancer cells. Carriers should be translocated inside blood stream to cancer cells, and be able to recognize and bind to characteristic receptor or other marker present on cancer cells.

Moreover, drug-carrier molecules after docking to cancer cells should be able to release anti-cancer agent(s) to cancer cell inducing cell death. This way of transport should increase natural compounds anti-cancer activity and simultaneously should reduce, or avoid potential side effects.

The reference list must be re-arranged for avoiding unnecesary repetitions

Author Response

Respond to review #2:

Thank you that you understand what should be done to help humans to live longer.

Additionally, I do not know how to explain that I attached the right manuscript file and currently in submission I can see previous version, I attach with this rebuttal letter the right version with name with Capital letters.

Reviewer 3 Report

This review gave an overview on the roles of natural compounds on preventive and therapeutic effects in cancer. Both the writing style and the content need major changes for this article to be useful to readers of the journal. 

Comments:

  1.        The authors should discuss the potential anti-metastasis’s effect of natural compounds in oral cancer.
  2.         Figure 2 has small labeling that may not be readable.
  3.         In Figure 3, the effect of molecular targets in different natural compounds (curcumin, resveratrol, lycopen, epigallocatechine-3-gallate, quercetin, acetogenin) should be separated in different Figures to make it easier to grasp.
  4.         The order and format of references used need to check.

Author Response

Respond to review #3:

  1. Thank you for your time and future’s directions what will be interesting to study in the future. The idea if natural compounds display some antimetastatic properties is really good and I will try to find out about it in the future. Currently, I think that data on this topic are limited. The most important is cancer prevention and stay healthy, without metastasis.
  2. All Figures are 333dpi, and I will change letters size if editor who will prepare volume will requested. It must be verified if it is necessary, because If I will change size of letters/symbols this will change for bigger the size of this Figure.
  3. Some peoples like tables, others separate Figures. This is really difficult to make everybody happy. From my point of view this Figures are enough informative.
  4. I attach a proper version of manuscript with references placed in good order. I do not know why the previous version is uploaded.

I understand that writing style could be a personal point of view, and some peoples like this review and others do not understand what includes. Thank you anyway for your time and valuable suggestions.

Round 2

Reviewer 1 Report

  1. Figure 2 is still unreadable. I recommend the author to print the article on a printer and see the quality of the illustration.
  2. I insist that the data should be formalized and presented in the form of tables. Solid text is difficult for the reader to perceive. Despite the fact that the authors want to publish the manuscript as soon as possible.
  3. Legends under the figures should be more informative

Author Response

Reviewer 3:

  1. The quality of the illustrations needs to be improved. Figures are hard to read
  2. The text of the review is written carelessly. Requires careful proofreading.
  3. Chapter titles should be more descriptive
  4. To improve the quality of the manuscript, tables should be built and additional illustrations prepared.
  5. References are designed carelessly and do not correspond to the rules of the journal

Response to Reviewer 3 suggestions:

  1. The quality of the illustrations needs to be improved. Figures are hard to read

I did send for publication a high quality Figures in digital quality. I think that editors should make them bigger if necessary. Everybody could have a different quality because of computers and printers quality. I will do something if requested by Editors.

  1. The text of the review is written carelessly. Requires careful proofreading.

Manuscript was 2 times corrected by specialized in English correction/translation office. First time before sending to publication, and after reviewers request for manuscript modification. I will upload corrected version with modification marked in yellow, corrected by Anthony Cosgrove, an author of books on English language teaching, so he must know English very well.

  1. Chapter titles should be more descriptive
  1. To improve the quality of the manuscript, tables should be built and additional illustrations prepared.

For both queries I would like to explain:

This is the same kind of suggestion like to make bigger number of tables, or discussion about metastasis. Everybody have a personal likes or dislikes. I hope that the scientific value of manuscript is correct. It is really difficult to make everybody happy.

  1. References are designed carelessly and do not correspond to the rules of the journal

I did prepare and sent manuscript with references like should be, but because manuscript was previously also in other places before upload, I do not know why not the last version of manuscript was upload. I did tried and check that I upload a correct one.

I hope that after my explanation manuscript will be suitable for publication.

After my several tries to publish this manuscript, and simultaneously also others, I have also learn a lot: I think that for speed manuscripts publication, as well as scientific development on the Earth we need a changes:

  1. There should be a kind of post office where all scientists should address their work (manuscripts) for publication
  2. On each continent should be offices (offices) collecting manuscripts for publication. In case of some complaining that should be also, a main headquarters for all globe should organize manuscripts publication.
  3. Manuscripts should be prepared according to one simple rules and prepared in template (template must be the same for all journals)
  4. In this office each manuscript should be characterised, based on abstract and key words (this kind of search already exist in Elsevier)
  5. After manuscript classification manuscript should be directed to proper journal and send for reviews.
  6. This way of reorganization will reduce waiting for manuscript publication (for this manuscript 5 years), will reduce cost of paper (ecology), used for preparing previous versions sent for publication. The most important value there is, that this way of manuscript publication will significantly save scientist’s time. Scientists will have time to do next experiment or for scientific discussions, or to read scientific papers. Therefore, one template will speed science development, reduce production of many versions of the same manuscript and loosing time for manuscript uploading for many journals.

From my point of view everybody will profit using this way of manuscript publication, as well as will speed the scientific development of science. Importantly, scientists will have more time for their families.

Reviewer 3 Report

I recommend to publish in this present form

Author Response

Response to Reviewer 2 suggestions:

  1. The authors should discuss the potential anti-metastasis’s effect of natural compounds in oral cancer.

 I did already more than 100 reviews, but in each review, if necessary
 I write what should be corrected, but in this case, there is not a merit query: of course I could discuss the influence of natural drugs in metastasis, but I did prepare the other manuscript about metastasis, where is a better place to discuss natural drugs involvement in metastasis.

  1. Figure 2 has small labeling that may not be readable.

I did send for publication a high quality Figures in digital quality. I think that editors should make them bigger if necessary. Everybody could have a different quality because of computers and printers quality. I will do something in this matter if will requested by Editors.

  1. In Figure 3, the effect of molecular targets in different natural compounds (curcumin, resveratrol, lycopen, epigallocatechine-3-gallate, quercetin, acetogenin) should be separated in different Figures to make it easier to grasp.

This is the same kind of suggestion like to make bigger number of tables. Everybody have a personal likes or dislikes and point of view. I hope that the scientific value of manuscript is correct. It is really difficult to make everybody happy.

  1. The order and format of references used need to check.

I did prepare and sent manuscript with references like should be prepared for this journal, but because of manuscript previous uploading in the other journals, I do not know how it has happened that not the last version of manuscript was upload. I did check that I upload a correct one.

Round 3

Reviewer 1 Report

My comments and wishes were practically not taken into account. Without tables, the text of the article is difficult for the reader to perceive.

Author Response

Reviewer 3: My comments and wishes were practically not taken into
account. Without tables, the text of the article is difficult for the
reader to perceive.

Respond to 2nd comments and to 3rd comments:

I would like to explain that this manuscript was previously sent to publication in Current Research in Biotechnology, where was reviewed and corrected as requested by reviewers. This manuscript was ready for publication, but was not publish, because of Current Research in Biotechnology had not enought Polish Ministry Points, and boss from my Department advice me to send this manuscript somwhere else, where I can have a ministry points. So I decide to upload manuscript to International Journal of Molecular Sciences. 

I undrestand that everybody have their own sense of taste, but because I am also a reviewer. and I did more that 100 reviews since year 2017 I know that if we have 3 reviewers, and 2 are happy and accept manuscript for publication, while a thirt one wants extra tables to be prepared. This is for person, who is also a reviewer,  not a merit mistake that could be a reason to block manuscript publication. Moreover, because of peoples different sens of tast, even if I agree and prepare tables, the other 2 reviewers could have a different opinions and comment addition of tables to this manuscript. It is usually difficult to meke everybody happy.

I hope that because of tables addition will not change anything in the merit of this manuscript, except better agent's order placed in the table, this form of manuscript will be suitable for publication. I hope that my explanation are enought and manuscript will be publish.

Thank you very much for your time and suggestions what should be changed.